# Secondary Cytoreductive Surgery in Relapsed Platinum-Sensitive Epithelial Ovarian Cancer: A Systematic Review of Randomized Controlled Trials

**DOI:** 10.3390/cancers16142613

**Published:** 2024-07-22

**Authors:** Andrea Svennevik Myhr, Line Bjørge, Cecilie Fredvik Torkildsen

**Affiliations:** 1Faculty of Medicine, University of Bergen, 5009 Bergen, Norway; 2Centre for Cancer Biomarkers, Department of Clinical Science, University of Bergen, 5009 Bergen, Norway; 3Department of Obstetrics and Gynecology, Haukeland University Hospital, 5009 Bergen, Norway; 4Department of Obstetrics and Gynecology, Stavanger University Hospital, 4068 Stavanger, Norway

**Keywords:** gynecologic cancer, ovarian cancer, oncology, epithelial ovarian cancer, secondary cytoreductive surgery

## Abstract

**Simple Summary:**

Secondary cytoreductive surgery is a treatment option for patients with relapsed platinum-sensitive epithelial ovarian cancer, yet the precise indications and criteria for patient selection remain to be outlined. Furthermore, the impact on progression-free and overall survival remains unclear. The objective of this systematic review was to determine the precise indications for secondary cytoreductive surgery and to elucidate the factors contributing to favorable outcomes associated with the intervention compared to conventional treatment modalities like standard-of-care chemotherapy. Our review confirmed that secondary cytoreductive surgery maintains morbidity, mortality, and quality of life standards for patients. While the trials included utilized different selection criteria for the procedure, our findings underscore the importance of careful patient selection to improve survival in conjunction with conventional chemotherapy.

**Abstract:**

Secondary cytoreductive surgery is a treatment option for relapsed platinum-sensitive epithelial ovarian cancer, but no clear indications are defined for the procedure. This systematic review aims to establish clear indications and compare outcomes versus standard-of-care chemotherapy. We conducted an electronic literature search across three databases and identified 2033 articles, including three phase 3 randomized controlled trials (RCT). The review adhered to PRISMA 2020 guidelines and was registered in PROSPERO (no. CRD42022379817). Despite varying patient selection methods, surgery plus chemotherapy demonstrated significantly prolonged progression-free survival compared to chemotherapy alone. However, overall survival outcomes were inconsistent: while GOG-0213 did not show extended overall survival, recent studies with stricter defined criteria for surgery (SOC-1 and DESKTOP-III) reported improved overall survival with the addition of surgery. Morbidity and mortality rates were low, with no difference in quality of life between the surgery and no-surgery groups. In conclusion, cytoreductive surgery presents a promising option for recurrent epithelial ovarian cancer treatment. Nonetheless, well-defined selection criteria appear crucial for achieving increased overall survival compared to conventional treatment.

## 1. Introduction

Epithelial ovarian cancer (EOC) is a lethal disease, with more than 60% of patients being diagnosed at advanced stages (International Federation of Gynecology and Obstetrics [FIGO] stages III–IV), and over 80% experiencing relapse and ultimately disease-related death [1,2,3]. Primary cytoreductive surgery (PCS) is well-established, but secondary cytoreductive surgery (SCS) at relapse has only recently gained recognition as a feasible option in meticulously chosen cases [4]. Studies published in the previous decades have presented divergent findings regarding the impact of SCS on overall survival (OS) [5,6,7]. However, more recent publications have shown encouraging outcomes, especially in preselected patient cohorts [8,9,10].

The two most important prognostic biomarkers for EOC are the cancer stage at diagnosis and the degree of cytoreduction in the primary setting. The positive effect of surgery in the primary treatment setting has been demonstrated in multiple trials, with the greatest benefit observed when complete cytoreduction is achieved [11,12,13,14]. While the value of cytoreduction in the neoadjuvant setting is more debated, most trials also emphasize the importance of complete surgery in this context [15,16,17]. Consequently, ultra-radical techniques have been developed to increase the rate of complete cytoreduction. Unfortunately, current procedures for assessing operability lack precision, a challenge that persists in evaluating the feasibility of extensive surgical interventions in cases of recurrent disease. Additionally, the role of tumor biology is gaining recognition as a factor that may impact survival irrespective of surgical effort [18].

Systemic therapy, in combination with surgery, plays a key role in the primary treatment setting. According to ESGO/ESMO guidelines, all patients with advanced disease should be offered chemotherapy. The anti-angiogenic agent bevacizumab is recommended to patients with a high risk of recurrence, based on the ICON-7 and the GOG-0218 trials, and for patients with platinum-sensitive homologous recombination deficiency (HRD) disease, based on the PAOLA-1 study [8,19,20]. Poly-ADP-ribose polymerase inhibitors (PARPi) have been widely implemented in the primary treatment setting due to the results from the PAOLA, SOLO1, and PRIMA trials [21,22,23]. Despite these improvements in primary treatment, a substantial number of women with advanced EOC still experience recurrence.

Treatment selection for patients with relapsed EOC requires multidisciplinary approaches. Most patients are treated with chemotherapy alone, ideally a platinum-based combination treatment (e.g., carboplatin and pegylated liposomal doxorubicin), followed by maintenance therapy with PARPis, olaparib, or niraparib (if not administrated as part of the first-line treatment regimen). In Norway, SCS is exclusively extended to carefully chosen patients for whom complete cytoreduction is anticipated [11], and the Norwegian National Guidelines for Gynecological Cancer advocate SCS solely in cases featuring platinum-sensitive, localized tumors without ascites [4,24].

The integration of SCS in relapsed EOC management is in the initial stages of investigation, and the potential effects on PFS, OS, and quality of life when compared to chemotherapy alone are currently unclear.

Objective guidelines for patient selection are yet to be established, leaving the preoperative assessment of SCS to local practices and personal experiences.

This systematic review (SR) will explore PFS, OS, and quality of life in patients with relapsed EOC who undergo SCS. It will also explore indications for SCS and compare models aimed at identifying patients most likely to benefit from surgical treatment.

## 2. Materials and Methods

### 2.1. Search Methods

This SR was performed in alignment with the PRISMA (Preferred Reporting Items for Systematic Reviews and Meta-Analyses) 2020 guidelines, and the review was registered in PROSPERO (No. CRD42022379817) [25]. The PICO (Population, Intervention, Comparison, and Outcomes) framework was used to identify keywords for the search strategy. On 6 November 2022, electronic databases Medline, Embase, and Cochrane were systematically explored for studies that provided data on treatment with secondary cytoreductive surgery in patients suffering from first relapse of platinum-sensitive epithelial ovarian cancer. The search was structured using variations of the keywords “secondary surgery”, “secondary debulking” and “secondary cytoreductive surgery” in combination with the term “epithelial ovarian cancer”. A complete list of the search algorithm is available in Appendix A. The search yielded a total of 2033 studies.

### 2.2. Study Design

Only studies published in English were included. No period limits were set. Only phase 3 RCTs comparing secondary cytoreductive surgery and chemotherapy to chemotherapy alone were included in this SR. Phase 1–2 studies, pilot studies, pooled analyses, case reposts, conference materials, summaries, reviews, and editorials were excluded. The final overall survival analysis of the SOC-1 randomized phase 3 trial was included separately at publication (post-literature search) as the reviewers considered it a continuation of the original SOC-1 trial [26].

### 2.3. Participants

Inclusion criteria for participants encompassed the following: adults >18 years of age; first recurrence of EOC; platinum-sensitive disease defined as a platinum-free interval (PFI) surpassing 12 months; and undergoing SCS concurrently with chemotherapy. PFI has long been a deciding factor in the treatment of recurrent epithelial ovarian cancer, but in more recent published data, it is suggested that PFI should only be used indicatively and as a part of a broader individual assessment [27]. Exclusion criteria were the following: comprised cases where SCS was not performed; studies involving tertiary or quaternary cytoreductive surgery (TCS/QCS); concurrent use of hyperthermic intraperitoneal chemotherapy (HIPEC); and studies authored by the same individual or their associates, involving an identical patient cohort. Tertiary and quaternary cytoreductive surgery are defined as, respectively, the third and fourth surgical attempts due to the second and third histologically proven epithelial ovarian cancer relapse. Hyperthermic intraperitoneal chemotherapy (HIPEC) is defined as a surgical procedure where heated chemotherapy is administered directly into the abdominal cavity, mainly offered in combination with cytoreductive surgery in the treatment of peritoneal carcinomatosis. The SOCceR trial was not included in the systematic review as it was prematurely ended [28].

### 2.4. Intervention

Only studies exploring SCS in combination with chemotherapy compared to chemotherapy alone were included. Additional treatment with angiogenic inhibitors or PARPi’s was allowed.

### 2.5. Research Questions

Which selection criteria most accurately predict the patients with relapsed EOC who will derive the most benefit from SCS?Does the addition of SCS to chemotherapy enhance PFS and OS in patients with relapsed EOC, compared to chemotherapy alone?Does SCS change the rate of morbidity and mortality compared to those treated solely with chemotherapy?Are the quality-of-life parameters influenced by treatment with both surgery and chemotherapy compared to only chemotherapy?

### 2.6. Outcome Measures

The primary objective was to define the appropriate indications for SCS. The assessment of efficacy also includes an evaluation of any survival benefit (OS and PFS) compared to chemotherapy alone. Important additional outcomes were differences in morbidity and mortality assessed by complication rates and mortality rates within a 30-day postoperative period, and quality of life measured using the Trial Outcome Index of the Functional Assessment of Cancer Therapy-Ovary (FACT-O TOI) and European Organization for Research and Treatment of Cancer (EORTC) 30-item core quality of life (QLQ-C30) questionnaires.

### 2.7. Study Selection

The articles were screened blindly by two participants (A.S.M. and C.F.T.) using the established criteria for inclusion and exclusion. Primarily, the title and abstracts were screened; secondly, a full-text read-through of all remaining studies was conducted. The semi-automated electronic screening tool Rayyan was used in the process [29]. Rayyan is “Software as a Service” (SaaS) that is subject to continuous updates and thus has no public version to reference. Any disagreements were ultimately resolved through discussion with a third reviewer (L.B.).

### 2.8. Data Collection

The extracted parameters included the total number of patients randomized to each treatment arm, rates of complete cytoreduction, data on OS and PFS for each treatment arm across the tree studies, documented morbidity and mortality rates, and evaluations of quality of life during treatment within each group.

### 2.9. Risk of Bias Assessment

In general, all surgical trials have a high risk of performance bias due to the open-label surgical treatment. Overall, GOG-0213, SOC-1, and Desktop III demonstrate a low risk of bias across most other domains.

### 2.10. Data Analysis

The research questions were formulated in accordance with the PICO framework. Estimated common effects are visualized using forest plots designed using Microsoft Excel. The combined values are estimated using weighted averages based on the total patient population in each study. All statistical analyses are purely descriptive.

## 3. Results

### 3.1. Study Selection

A total of 2033 studies were identified across three online databases (Medline, Embase, and Cochrane). Before initiating the screening process, 832 duplicates were removed between the two software programs EndNote and Rayyan, leaving 1201 studies to be screened by reading headlines and abstracts. Of these, 23 studies were reviewed in full text, with three phase 3 RCTs ultimately being included in the SR. The search included articles published between 1946 and 4 November 2022. Details of the selection process can be obtained from the PRISMA flow diagram in Figure 1.

### 3.2. Study Characteristics

Three phase 3 RCTs, representing 1249 patients, were ultimately included in this SR. A detailed overview of the characteristics of the trials is presented in Table 1. All trials included participants aged 18 years or older at their first epithelial ovarian cancer relapse after a PFI of 6 months or more. Selection criteria for surgery differed between the studies, but all trials aimed for an equal distribution of patients between the surgery and no-surgery groups. Data regarding PFS and OS were reported in each of the trials. However, at data closure, the OS data from the SOC-1 trial were not reported.

### 3.3. Progression-Free Survival (PFS) and Overall Survival (OS)

GOG-0213 found the PFS of patients in the surgery group who underwent SCS to be longer than patients in the no surgery group, reporting a median PFS of 18.9 months in the surgery group compared to a median PFS of 16.2 months in the no surgery group (HR 0.82, 95% CI, 0.66–1.01). However, GOG-0213 did not find any OS benefit associated with SCS and concluded that SCS in combination with chemotherapy did not result in longer OS than chemotherapy alone. The median OS in the surgery group was 50.6 months compared to 64.7 months in the no-surgery group (HR 1.29, 95% CI, 0.97–1.72).

SOC-1 found an increased PFS in patients randomized to the surgery group who underwent SCS in combination with chemotherapy, compared to the no-surgery group. The median PFS was 17.4 months in the surgery group compared to 11.9 months in the no-surgery group (HR 0.58, 95% CI, 0.45–0.74). Mature OS data were recently reported in the final overall survival analysis of the SOC-1 randomized phase 3 trial. The median OS was 58.1 months in the surgery group and 52.1 months in the no-surgery group (HR 0.80, 95% CI, 0.61–1.05), all though the threshold for statistical significance was not met (*p* = 0.11). After adjusting for the 61 (35%) patients in the no-surgery group who crossed over to the surgery group following relapse, HR for death in the surgery group compared with the no-surgery group was 0.76 (95% CI, 0.58–0.99). The most favorable outcomes were reported in patients who underwent surgery with complete resection (73.0 months median OS with CR, compared to 52.1 months in the no-surgery group). A prolonged OS was also observed in the subgroup of patients with <20 sites of relapse (not estimable in the surgery group, 69.5 months in the no-surgery group).

Corresponding to the results of SOC-1, DESKTOP-III reported both a PFS and OS benefit associated with SCS. The median PFS was 18.4 months in the surgery group and 14.0 months in the no-surgery group (HR 0.66, 95% CI, 0.54–0.82). The median OS was 53.7 months in the surgery group and 46,0 months in the no surgery group (HR 0.75, 95% CI, 0.59–0.96). Also consistent with the results of SOC-1, DESKTOP-III reported a particularly prolonged OS in patients who underwent SCS where CR was achieved (61.9 months median OS with CR, compared to 27.7 months in the no-surgery group).

Forest plots illustrating survival data across the three studies are given in Figure 2, Figure 3, Figure 4 and Figure 5. A clustered bar chart displaying PFS and OS across the three RCTs is given in Appendix B.

### 3.4. Selection Criteria

The three studies identified employed different models for assessing the probability of achieving complete resection during SCS. An overview of the selection criteria applied in the three studies is given in Table 2. In the GOG-0213 study, patients were deemed eligible for surgery if they had investigator-determined resectable disease [8]. Patients were also required to have adequate renal, hepatic, and bone marrow function, as well as a GOG performance-status score of 0 to 2. Women with diffuse carcinomatosis, ascites, or extra-abdominal disease were excluded.

The SOC study used the iMODEL score in combination with PET-CT imaging to determine which patients would be most likely to benefit from SCS. The iMODEL score, a recently formulated model for patient selection, is computed based on six variables, including FIGO stage, residual disease, PFI, ECOG score, CA125 level at recurrence, and ascitic fluid [9].

For patient selection, DESKTOP-III used the AGO score previously presented and validated through the DESKTOP-I and DESKTOP-II studies [11,30]. Patients were eligible for surgery if they presented with a positive AGO score (ECOG 0, ascitic fluid < 500 mL, and primary cytoreductive surgery without residual tumor). Primary cytoreductive surgery (PCS) is defined as the first surgical attempt to treat histologically proven ovarian cancer) [8].

The rates of CR varied among the studies. In GOG-0213, CR was achieved in 67% of the patients assigned to surgery who underwent the procedure. In comparison, SOC-1 and DESKTOP-III reported CR rates of 77% and 75.5%, respectively.

### 3.5. Morbidity and Mortality

The GOG-0213 trial reported one patient dying from post-operative complications (pulmonary embolism), corresponding to a mortality rate of 0.4%. Two patients in the no-surgery group died of cardiopulmonary arrest and were considered by the investigator to be “at least possibly” related to trial treatment. At 30 days after treatment, the surgical morbidity was reported to 9% (20 patients).

The SOC-1 trial reported no patient death in either group at 60 days after receiving assigned treatment. Postoperative complications up to 30 days after surgery were graded using the Memorial Sloan-Kettering Cancer complication severity grading method. In the surgery group, nine (5%) of 172 patients had grade 3 or worse morbidity at 30 days. The most common grade 3–4 adverse events during chemotherapy were neutropenia (29 [17%] of 166 patients in the surgery group vs. 19 [12%] in the no surgery group), leucopenia (14 [8%] vs. eight [5%]), and anemia (ten [6%] vs. 9 [6%]). Four serious adverse events occurred, all in the surgery group: neutropenia (3 patients [2%]) and compromised kidney function (one patient [1%]).

In the DESKTOP-III trial, there was no reported perioperative mortality within 30 days after surgery. The rate of surgical morbidity was not provided.

### 3.6. Quality of Life

Assessments of quality of life were reported in each of the trials. The GOG-0213 study assessed patient-reported quality of life using the Trial Outcome Index of the Functional Assessment of Cancer Therapy-Ovary (FACT-O TOI) score. Surgery was associated with significantly lower postoperative FACT-O TOI scores. The scores were similar in the two groups at 6 weeks after treatment and at every subsequent assessment.

The SOC-213 trial assessed patient-reported quality of life using the European Organization for Research and Treatment of Cancer (EORTC) 30-item core quality of life questionnaire (QLQ-C30; global health status with score 0–100) and FACT-O TOI score. All patients were assessed at baseline and at 6, 12, 24, and 60 months after randomization. During the substantial follow-up period, there were at no point reported any significant differences in patient-reported quality of life between the surgery and no-surgery groups.

The DESKTOP-III study used the following tools for the assessment of patient-reported quality of life: EORTC QLQ-C30 and FACT-O TOI scores. All patients were assessed at baseline, at 6 months, and at 12 months after randomization. The results did not reveal any statistically significant differences between the groups with regards to global health status, quality of life, or any functional subscale at any of the three follow-ups. Model-based estimates of the between-group difference in the changes from baseline to 6 months indicated symptoms of insomnia and constipation more frequently in the surgery group. However, at this point in the treatment, 32 of 85 patients (38%) in the surgery group were still receiving chemotherapy, compared to only 11 of 99 patients (11%) in the no-surgery group. There were no observed between-group differences regarding these, or any other symptoms, at the 12-month evaluation.

## 4. Discussion

Ovarian cancer remains the most lethal gynecologic malignancy, often diagnosed at advanced stages with a high risk of recurrence. While primary cytoreductive surgery is widely practiced, secondary cytoreductive surgery has emerged as a novel option that can lead to improved PFS, but the effect on OS is still debated [4]. Despite varied conclusions among the three RCTs reviewed, multiple non-randomized studies suggest a survival advantage with SCS [7,31,32,33]. A 2022 meta-analysis of 36 studies encompassing 2805 patients with platinum-sensitive recurrent epithelial ovarian cancer similarly found that SCS, particularly when combined with maximal tumor resection, significantly prolongs OS [7].

While the three RCTs employed various methods to identify suitable candidates for SCS, they all underscore the importance of careful patient selection, prioritizing the likelihood of achieving complete resection for securing a survival advantage from SCS. Overall, SCS emerges as a promising treatment option for patients with recurrent EOC. However, the identification of precise predictive biomarkers for treatment response remains a matter of debate. Relapse of OC after treatment with PARPi represents a new clinical challenge. Neither the iMODEL nor the AGO score incorporate BRCA/HRD status or maintenance therapy with PARPi. Consequently, forecasting the applicability of findings from the SOC-1 and DESKTOP III trials poses a challenge. The emergence of robust data from trials investigating SCS in patients undergoing PARPi treatment is awaited to address this gap [34].

Survival is the strongest indicator of benefit for patients undergoing SCS. Across the studies, SCS in combination with chemotherapy was associated with a significantly longer PFS than systemic therapy alone. However, the findings regarding overall survival were incongruent. DESKTOP-III found cytoreductive surgery followed by chemotherapy resulted in a longer OS than chemotherapy alone. Mature data from SOC-1 did not find an overall OS benefit in SCS compared to chemotherapy alone; however, after adjusting for patients in the no-surgery group who crossed over to the surgery group following relapse, HR for death was 0.76 (surgery vs. no-surgery), a comparable result to the OS data of DESKTOP-III. The results of the GOG-0213 study stand out with the highest three-year OS rate in the no-surgery arm (GOG-0213 75%, SOC-1 66%, and DESKTOP-III 62%) and the lowest OS in the arm undergoing surgery (GOG-0213 76%, SOC-1 78%, and DESKTOP-III 84%) among the three studies. As previous SR and meta-analyses have suggested, the difference in outcomes between the studies may have been a result of the implementation of different selection criteria (SOC-1 using iMODEL score and DESKTOP using AGO score) or the absence of standardized selection criteria (GOG-0213) [31,35].

Survival could also be influenced by differences in the patient populations, like variation in disease burden, implementation of anti-angiogenetic treatment as standard, and time frame from SCS to initiation of postoperative chemotherapy. While >2/3 of the patient population in SOC-1 and DESKTOP-III suffered from multifocal disease, including peritoneal carcinomatosis, most women with peritoneal carcinomatosis were excluded from GOG-0213 as they were considered poor candidates for surgery. This raises an important issue: the significance of surgical quality and the role of highly specialized ovarian cancer surgical centers. The maintenance of high surgical standards has become increasingly critical. Notably, in both the DESKTOP-III and SOC-1 trials, only specialized high-volume centers were permitted to enroll patients. This factor likely influences the proportion of patients eligible for secondary cytoreductive surgery and, consequently, their survival outcomes. Furthermore, 84% of patients in GOG-0213 received bevacizumab as part of the chemotherapy regimen, compared to 23% in the DESKTOP III trial and 1% in the SOC-1 trial. An abundance of bevacizumab administration may have contributed to an unexpectedly favorable performance within the chemotherapy-alone group, potentially attenuating the relative benefit observed in the intervention arm [7]. In addition to VEGF inhibitors, PARPi maintenance therapy was allowed in SOC-1 and DESKTOP-III, with 12% of the patients in SOC-1 and 5% of the patients in DESKTOP-III receiving PARPis as part of the treatment regimen. GOG-0213 did not provide any information regarding PARPi use in their population. A recently published 2024 retrospective cohort study found reduced efficacy of SCS in patients previously treated with PARPis compared to those without prior PARPi treatment, although further trials are necessary to define the roles of both bevacizumab and PARPis in combination with SCS [36]. The median time from surgery to subsequent chemotherapy was 40 days in GOG-0213, compared to 35 days in DESKTOP-III and 16 days in SOC-1. A post hoc analysis of the GOG-0213 study concluded time from surgery to chemotherapy initiation was predictive of OS, with the complete resection group encountering an increased risk of death when the time to initiation of chemotherapy exceeded 25 days, implying a shorter interval from surgery to chemotherapy may be beneficial [37].

Morbidity and mortality rates are crucial in assessing whether SCS should be widely offered to patients with relapsed epithelial ovarian cancer. The rates were assessed in the surgery and no-surgery arms of the three trials. Survival data were consistent, reporting only one death within 30 days after surgery across a total of 613 patients randomized for surgery between the three studies. There were no reported patient deaths in any of the no-surgery arms. Assessment of patient morbidity proved more challenging, as only SOC-1 provided an extensive list of post-operative complications. In total, the postoperative morbidity rates were relatively low (5% and 9% for SOC-1 and GOG-0213, respectively; data not provided in DESKTOP-III), especially considering the ultraradical surgical techniques used and the fact that neither of the standardized models (iMODEL, AGO) included surgical ability as a parameter for achieving complete cytoreduction.

Although these trials have investigated secondary cytoreductive surgery for patients experiencing recurrence of epithelial ovarian cancer, there is currently no consistent method to identify the best candidates for this intervention. All trials include ECOG performance status and platinum-free interval of at least six months in their model. Both the iMODEL and AGO scores utilize the volume of ascitic fluid at recurrence. GOG-0213 relied on a complete response to front-line therapy, whereas IMODEL and AGO use complete primary surgery in frontline treatment. In GOG-0213, the surgical decision was left to the investigator’s discretion. This highlights the challenge of establishing a reliable biomarker or model for secondary cytoreductive surgery, echoing the complexities also observed in primary cytoreductive surgery. Preoperative assessment in primary cytoreductive surgery relies on diverse imaging modalities, laparoscopic techniques, and phenotypic risk scores, either individually or in combination. Treatment institutions have developed algorithms based on local protocols and personal experiences, yet none have emerged as uniformly superior. It is also important to note that among patients with negative AGO scores, the likelihood of achieving complete secondary resection remains at 50%. While this scoring system effectively reduces unnecessary procedures, it may inadvertently underestimate candidates for SCS, potentially compromising patient care. Furthermore, existing stratification methods often overlook the biological heterogeneity inherent in epithelial ovarian cancer. Addressing this diversity is essential for refining patient selection and optimizing treatment outcomes.

This systematic review is strengthened by its comprehensive search and synthesis of prior research on secondary cytoreductive surgery in epithelial ovarian cancer, including recent survival data from the SOC-1 trial. Despite its acceptance, only three randomized controlled trials have investigated this procedure, each employing different methodologies and reporting varied outcomes. Our manuscript also highlights the strengths and weaknesses of these similar but different trials. Our findings suggest that while secondary cytoreductive surgery is recommended for select patients with relapsed epithelial ovarian cancer, uncertainty remains in patient selection criteria, posing a limitation to this review. The existing trials differ significantly in inclusion criteria, scoring systems, surgical techniques, and use of maintenance therapy.

The results from this review are aligned with those of previous SRs [38,39].

## 5. Conclusions

In conclusion, this SR suggests patients with relapsed platinum-sensitive epithelial ovarian cancer may benefit from complete or optimal cytoreductive surgery and thus should be evaluated for surgical treatment in the context of multidisciplinary team decision making based on their functional status and disease burden. Results were conflicting concerning a possible survival benefit in SCS. However, the combined results of SOC-1 and DESKTOP-III align well with previously published studies. The three trials were undivided in reporting a low risk of mortality and morbidity and a transient decrease in patient-reported quality of life in the postoperative period. Further prospective randomized controlled trials are essential to refining the selection criteria for secondary cytoreductive surgery. Additionally, continued exploration into the role of earlier maintenance therapy and the benefits of secondary cytoreductive surgery in these patients is necessary. Initial findings from ongoing trials addressing these aspects are expected by 2028.

## Figures and Tables

**Figure 1 cancers-16-02613-f001:**
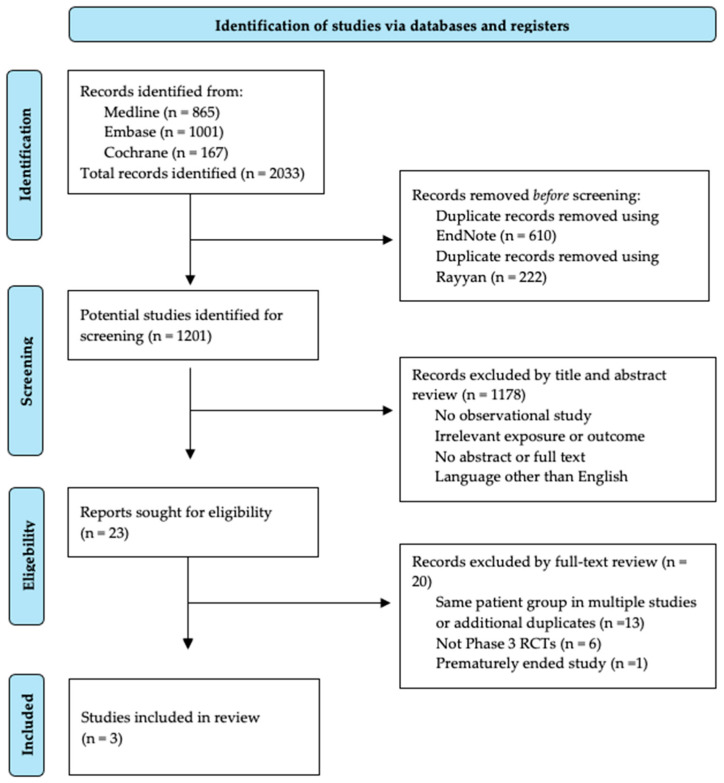
PRISMA flowchart of study selection for systematic review protocol [25].

**Figure 2 cancers-16-02613-f002:**
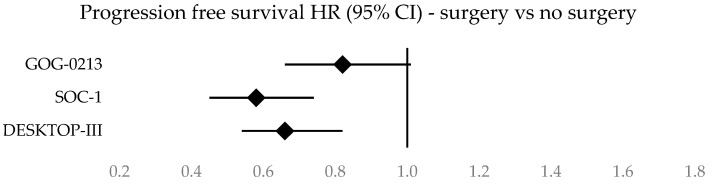
Forest plot of progression-free survival of patients in the surgery vs. no-surgery groups.

**Figure 3 cancers-16-02613-f003:**
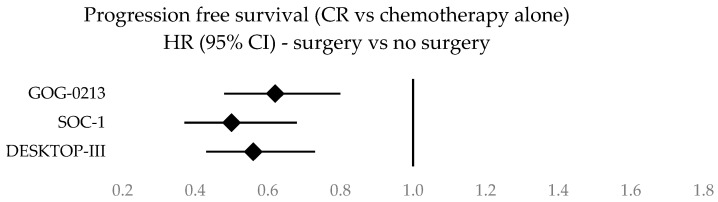
Forest plot of progression-free survival in patients with SCS where complete resection is achieved vs. chemotherapy alone. CR: complete resection.

**Figure 4 cancers-16-02613-f004:**
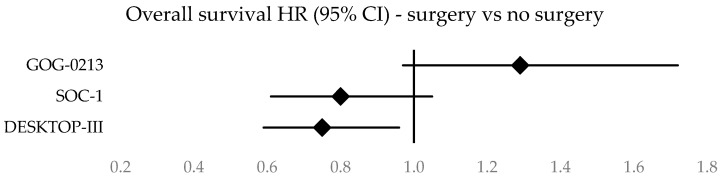
Forest plot of overall survival.

**Figure 5 cancers-16-02613-f005:**
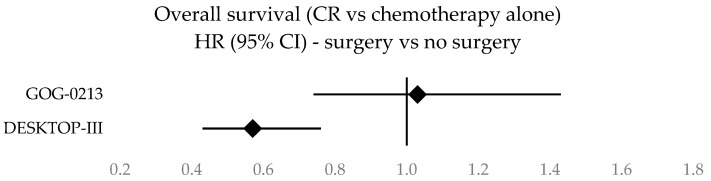
Forest plot of overall survival in patients with SCS where complete resection is achieved vs. chemotherapy alone. CR: complete resection.

**Table 1 cancers-16-02613-t001:** Studies characteristics. SCS: secondary cytoreductive surgery; CR: complete resection; OS: overall survival; PFS: progression-free survival; AGO: Die Arbeitsgemeinschaft Gynäkologische Onkologie.

Author	Name of Study	Year of Publication	Total Patients Included (n)	Patients Treated with SCS (n)	Rate of CR (%)	Patients Treated with Chemotherapy Alone (n)	Selection Criteria	Primary End Point	Median Outcomes SCS (Months)	Median Outcomes Chemotherapy (Months)
Coleman et al. [8]	GOG-0213	2019	485	240	67	245	Investigator determined	OS	OS 50.6 PFS 18.9	OS 64.7 PFS 16.2
Zang et al. [9]	SOC-1	2020	357	182	76.7	175	iMODEL	OS and PFS	OS 58.1 PFS 17.4	OS 52.1 PFS 11.9
Harter et al. [10]	DESKTOP-III	2020	407	206	74.5	201	AGO-score	OS	OS 53.7 PFS 18.4	OS 46.0 PFS 14.0

**Table 2 cancers-16-02613-t002:** A systematic display comparing the three selection criteria implemented in GOG-0213, DESKTOP-III and SOC-1. ECOG: Eastern Cooperative Oncology Group; FIGO: International Federation of Gynecology and Obstetrics. ^a^: Patients who fail to meet this specific criterion may be eligible for SCS if their total iMODEL score is 4.7 or lower out of 11.9. A lower total score suggests a higher likelihood of achieving complete resection.

	Investigator Determined	AGO-Score	iMODEL Score
Investigator determined resectable disease	Yes	No	No
ECOG Performance status (0–5)	No	Yes, ECOG 0	Yes, ECOG 0 or 1 ^a^
Limitations to ascites at recurrence	No	Yes, ≤500 mL	Yes, 0 mL ^a^
Macroscopically complete resection at primary cytoreductive surgery (PCS)	No	Yes	Yes ^a^
Limitations to CA125 at recurrence	No	No	Yes, <105 U/mL ^a^
FIGO stage (0–4)	No	No	Yes, FIGO stage 1 or 2 ^a^
Platinum-free interval	Yes, ≥6 months (platinum-sensitive disease)	Yes, ≥6 months (platinum-sensitive disease)	Yes, ≥16 months (platinum-sensitive disease) ^a^
PET-CT	No	No	Yes, but not considered a part of the original iMODEL score

## Data Availability

The data presented in this study are available in this article.

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
