# Peer review of "Secondary Cytoreductive Surgery in Relapsed Platinum-Sensitive Epithelial Ovarian Cancer: A Systematic Review of Randomized Controlled Trials"

_cancers, 2024, doi:10.3390/cancers16142613_

Round 1
Reviewer 1 Report
Comments and Suggestions for Authors
There are only three studies included in the review. Additionally, I noticed that the author only searched for published studies between 1946 and November 4th, 2022. The author should update the literature search to include more recent studies.
Author Response
#1 There are only three studies included in the review. Additionally, I noticed that the author only searched for published studies between 1946 and November 4th, 2022. The author should update the literature search to include more recent studies.
Reply:
The data presented in this manuscript are derived from the MD thesis “Secondary cytoreductive surgery in relapsed platinum-sensitive ovarian cancer: A systematic review” submitted to and approved by the University of Bergen in Fall 2023. An extensive literature search conducted in Spring 2023 did not reveal any new randomized phase III trials. The only additional relevant publication identified was the final overall survival analysis of the SOC-1 trial. This finding has been corroborated during the manuscript review process.
Per request of the Reviewer, an updated literature search was conducted July 9th 2024 employing the same search strategy as in the original search, but restricted to articles published between 2022 and July 9th 2024. The search yielded the following hits:
Ovid Medline: A total of 103 articles. The only RCT identified was the trial below, which is expected to be completed by 2026.
https://ovidsp.dc1.ovid.com/ovid-new-b/ovidweb.cgi?&S=BLOGFPIJPCACCHIFKPJJLGPMGJKOAA00&Complete+Reference=S.sh.55%7c22%7c1&Counter5=SS_view_found_complete%7c37945326%7cppez%7cmedline%7cmedl&Counter5Data=37945326%7cppez%7cmedline%7cmedl
Embase: A total of 138 articles. The search identified the same trial as in Medline, as well as the trial below, which is also expected to be completed by 2026.
https://ovidsp.dc1.ovid.com/ovid-new-b/ovidweb.cgi?&S=FNMCFPGJDPACCHGMKPJJAHPKILAPAA00&Complete+Reference=S.sh.65%7c4%7c1&Counter5=SS_view_found_complete%7c2029561665%7coemezd%7cembase%7cemexb&Counter5Data=2029561665%7coemezd%7cembase%7cemexb
Cochrane: A total of 24 articles. The same articles which were found in Medline and Embase were also identified in Cochrane, as well as the following trial which focuses on SCS in relapsed ovarian cancer after PARPi maintenance treatment, with expected completion by 2028.
https://www.cochranelibrary.com/central/doi/10.1002/central/CN-02495203/full
In conclusion, no new RCTs have been published in the period November 2022 - July 2024. Following internal discussions, we have decided to present only the original data.

Reviewer 2 Report
Comments and Suggestions for Authors
I have carefully reviewed "Secondary Cytoreductive Surgery in Relapsed Platinum-Sensitive Ovarian Cancer: A Systematic Review," which is well structured and well written.
My comments and suggestions:
INTRODUCTION:
Page 2, line 50: Suggest using "epithelial ovarian cancer" throughout the text instead of just "ovarian cancer," as the article deals with this specific group.
Page 2, line 50: It is necessary to add that the value of surgery has been questioned in randomized studies with neoadjuvant therapy and that there is no absolute truth about the value of primary cytoreduction, contrary to what the authors imply.
Page 2, line 60: Add information about the recent study with Niraparib by Antonio González-Martín (NEJM). Although the authors cite the study later, they do not describe it.
Page 2, line 88: This is the first time the abbreviation “SR" is used. The full term should be provided before the first use of the abbreviation.
RESULTS:
Page 4, line 169: While I understand the term “citostatic," I believe it would be better to use the standard term “chemotherapy.”
Page 4, line 176: The authors have already mentioned in the methods that the search was conducted according to the PRISMA protocol; the first sentence can be omitted. The second and third sentences can be repositioned to follow "A total of 2033 studies were identified across three online databases (Medline, Embase, and Cochrane)” for better flow.
Page 5, line 192: To clarify the last sentences, I suggest modifying it to "Data regarding PFS and OS were reported in each of the trials. However, at data closure, the OS data from the SOC-1 trial were not reported."
Page 5, line 201: It seems that the text data do not match the Forest plot graphics. Notice that the disease-free survival of GOG 213 cited has HR 0.82, 95% CI, 0.66-1.01. In the graphic, the horizontal line through the squares indicates the confidence intervals (CI) of the study, and according to the graphic, it would be more or less 0.2 - 1.8. The confidence interval information in the graphics seems to differ from the text. I suggest reviewing the Forest Plot graphics, but I am not an expert in statistics, so I may be wrong.
Page 7: I suggest including in the graphics a diamond representing the combined effect of all studies in the meta-analysis. The center point of the diamond shows the combined estimate of the effect, and the ends of the diamond represent the confidence interval of that combined estimate.
Page 8, line 244: There is a formatting error, as the sentence is split.
Page 10, line 312: "Three RCTs investigating the effect of SCS on survival, morbidity/mortality, and quality of life across 1249 patients with recurrent EOC were identified.” This data can be presented in the results. In the discussion, it is an unnecessary repetition.
Page 10, line 319: I believe it is important to reflect on the differences between the centers that recruited patients in the three studies. It seems that the European and Chinese studies included only centers with recognized expertise in cytoreduction, while GOG2013 does not appear to have had any minimum requirement regarding the surgical qualification of the center. This is an increasingly frequent discussion, and the need for cytoreductions to be performed in certified reference centers has been emphasized.
Page 10, line 345: Suggest changing “angiogenetic" to “anti-angiogenic.”
CONCLUSION:
Page 11, lines 405 to 410 are essentially a repetition of the discussion. I suggest removing them.
I suggest removing the references from the conclusion and starting it with the sentence that is currently at the end of the conclusion. Then, the authors can suggest future directions and research that can be followed.
Author Response
We appreciate the reviewers' thorough evaluation and constructive comments on our manuscript, titled “Secondary cytoreductive surgery in relapsed platinum-sensitive ovarian cancer: A systematic review” (Cancers-3082260). We are grateful for the opportunity to revise and resubmit our work. Below, we provide detailed, point-by-point responses to the reviewer’s questions and comments.
REVIEWER 2
INTRODUCTION:
#1 Page 2, line 50: Suggest using "epithelial ovarian cancer" throughout the text instead of just "ovarian cancer," as the article deals with this specific group.
Reply
Thank you for your suggestion. This has been incorporated throughout the manuscript
#2 Page 2, line 50: It is necessary to add that the value of surgery has been questioned in randomized studies with neoadjuvant therapy and that there is no absolute truth about the value of primary cytoreduction, contrary to what the authors imply.
Reply
Thank you for pointing out this important issue. The sentence is a simplification and could be misinterpreted. The text has been changed:
“The positive effect of surgery in the primary treatment setting has been demonstrated in multiple trials, with the greatest benefit observed when complete cytoreduction is achieved (11, 12, 13, 14). While the value of cytoreduction in the neoadjuvant setting is more debated, most trials also emphasize the importance of complete surgery in this context”.
#3 Page 2, line 60: Add information about the recent study with Niraparib by Antonio González-Martín (NEJM: https://www.nejm.org/doi/full/10.1056/NEJMoa1910962). Although the authors cite the study later, they do not describe it.
Reply
Thank you for highlighting this important trial examining the effect of Niraparib in first line treatment of all patients with EOC both with and without HRD. However, our review primarily focuses on secondary cytoreductive surgery rather than primary treatment. Therefore, including specific data from the PRIMA study, does not add significant value to this manuscript.
#4 Page 2, line 88: This is the first time the abbreviation “SR" is used. The full term should be provided before the first use of the abbreviation.
Reply
Thank you for pointing this out. The full term has now been provided upon its first occurrence.
RESULTS:
#5 Page 4, line 169: While I understand the term “citostatic," I believe it would be better to use the standard term “chemotherapy.”
Reply
Thank you for your suggestion, which has enhanced the clarity of the manuscript. We have Incorporated this improvement.
#6 Page 4, line 176: The authors have already mentioned in the methods that the search was conducted according to the PRISMA protocol; the first sentence can be omitted. The second and third sentences can be repositioned to follow "A total of 2033 studies were identified across three online databases (Medline, Embase, and Cochrane)” for better flow.
Reply
We appreciate the suggestion and agree that repositioning the sentences improves the manuscript’s flow. The suggested changes have been incorporated.
#7 Page 5, line 192: To clarify the last sentences, I suggest modifying it to "Data regarding PFS and OS were reported in each of the trials. However, at data closure, the OS data from the SOC-1 trial were not reported."
Reply
Thank you for this revision. It has been integrated.
#8 Page 5, line 201: It seems that the text data do not match the Forest plot graphics. Notice that the disease-free survival of GOG 213 cited has HR 0.82, 95% CI, 0.66-1.01. In the graphic, the horizontal line through the squares indicates the confidence intervals (CI) of the study, and according to the graphic, it would be more or less 0.2 - 1.8. The confidence interval information in the graphics seems to differ from the text. I suggest reviewing the Forest Plot graphics, but I am not an expert in statistics, so I may be wrong.
Reply
Thank you for pointing this out. The errors in the Figures 2-5 have been corrected and updated versions are included in the revised manuscript.
#9 Page 7: I suggest including in the graphics a diamond representing the combined effect of all studies in the meta-analysis. The center point of the diamond shows the combined estimate of the effect, and the ends of the diamond represent the confidence interval of that combined estimate.
Reply
This study constitutes a systematic review rather than a meta-analysis. The presented data are treated independently, and forest plots are utilized solely for visual representation of the included trials.
#10 Page 8, line 244: There is a formatting error, as the sentence is split.
Reply
Thank you for noticing this formatting error. This is now corrected.
#11 Page 10, line 312: "Three RCTs investigating the effect of SCS on survival, morbidity/mortality, and quality of life across 1249 patients with recurrent EOC were identified.” This data can be presented in the results. In the discussion, it is an unnecessary repetition.
Reply
We agree with the Reviewer and the sentence is now removed from the manuscript.
#12 Page 10, line 319: I believe it is important to reflect on the differences between the centers that recruited patients in the three studies. It seems that the European and Chinese studies included only centers with recognized expertise in cytoreduction, while GOG2013 does not appear to have had any minimum requirement regarding the surgical qualification of the center. This is an increasingly frequent discussion, and the need for cytoreductions to be performed in certified reference centers has been emphasized.
Reply
Thank you for raising this highly relevant topic. This is a weakness with the GOG0213 trial, and a strength with the two other trials. We paragraph addressing this topic has been included in the discussion:
“This raises an important issue: the significance of surgical quality and the role of highly specialized ovarian cancer surgical centers. The maintenance of high surgical standards has become increasingly critical. Notably in both the DESKTOP-III and SOC-1 trials only specialized high-volume centers were permitted to enroll patients. This factor likely influences the proportion of patients eligible for secondary cytoreductive surgery and consequently their survival outcomes.”
#13 Page 10, line 345: Suggest changing “angiogenetic" to “anti-angiogenic.”
Reply
Thank you for pointing out this mistake. The term is now corrected.
CONCLUSION:
#13 Page 11, lines 405 to 410 are essentially a repetition of the discussion. I suggest removing them.
I suggest removing the references from the conclusion and starting it with the sentence that is currently at the end of the conclusion. Then, the authors can suggest future directions and research that can be followed.
Reply
Thank you for this suggestion. Lines 405 to 410 are now removed from the conclusion. The last sentence is now moved to the beginning and references are removed from the conclusion. To address future directions and research that can be followed the following text is also added to the conclusion:
Further prospective randomized controlled trials are essential to refine the selection criteria for secondary cytoreductive surgery. Additionally, continued exploration into the role of earlier maintenance therapy and the benefits of secondary cytoreductive surgery in these patients is necessary. Initial findings from ongoing trials addressing these aspects are expected by 2028.

Reviewer 3 Report
Comments and Suggestions for Authors
The authors present the outcomes from a systematic review of the literature aiming to evaluate PFS, OS, and quality of life in patients with relapsed EOC who undergo secondary cytoreductive surgery. This is an interesting and somewhat well structured systematic review, however a number of issues need to be addressed:
- The title needs to be revised to reflect inclusion of only RCTs
- The introduction section is way too long and needs to be revised
- L109 "The 109 SOCceR trial was not included..." This needs to move to the included excluded section in the results
- Research questions belong to the methods section not the results
- L192 this sentence doesn't make sense please revise
- The strengths and limitations section needs to be revised. Large number of studies screened blindly by 2 reviewers is clearly not a strength of a SR, it is a part of the standard procedure. Please remove and add the limitations better
- I believe it should be highlighted in the conclusion section that SCS should be provided in the context of MDT decision making as previously mentioned in the discussion section
Comments on the Quality of English LanguageEnglish language could be improved with minor corrections throughout the manuscript
Author Response
We appreciate the reviewers' thorough evaluation and constructive comments on our manuscript, titled “Secondary cytoreductive surgery in relapsed platinum-sensitive ovarian cancer: A systematic review” (Cancers-3082260). We are grateful for the opportunity to revise and resubmit our work. Below, we provide detailed, point-by-point responses to the reviewer’s questions and comments.
REVIEWER 3
The authors present the outcomes from a systematic review of the literature aiming to evaluate PFS, OS, and quality of life in patients with relapsed EOC who undergo secondary cytoreductive surgery. This is an interesting and somewhat well structured systematic review, however a number of issues need to be addressed:
#1 The title needs to be revised to reflect inclusion of only RCTs
Reply
We appreciate your suggestion. The title is now changed to “Secondary Cytoreductive Surgery in Relapsed Platinum-Sensitive Epithelial Ovarian Cancer: A Systematic Review of Randomized Controlled Trials”
#2 The introduction section is way too long and needs to be revised.
Reply
Following your suggestion, we have reduced the introduction section by approximately 20%.
#3 L109 "The 109 SOCceR trial was not included..." This needs to move to the included excluded section in the results.
Reply
Thank you for this suggestion. The sentence is now moved to the section 2.3. Participants.
#4 Research questions belong to the methods section not the results.
Reply
Thank you for pointing out this mistake. The research questions are now moved from Results to Methods.
#5 L192 this sentence doesn't make sense please revise
Reply
We appreciate your observation. The sentence is now revised:
However, at data closure, the OS data from the SOC-1 trial were not reported.
#6 The strengths and limitations section needs to be revised. Large number of studies screened blindly by 2 reviewers is clearly not a strength of a SR, it is a part of the standard procedure. Please remove and add the limitations better.
Reply
We agree that the strength and limitations of the review needs to be revised. The following text is added:
“This systematic review is strengthened by its comprehensive search and synthesis of prior research on secondary cytoreductive surgery in epithelial ovarian cancer, including recent survival data from the SOC-1 trial. Despite its acceptance, only three randomized controlled trials have investigated this procedure, each employing different methodologies and reporting varied outcomes. Our manuscript also highlights the strengths and weaknesses of these similar but different trials. Our findings suggest that while secondary cytoreductive surgery is recommended for select patients with relapsed epithelial ovarian cancer, uncertainty remains in patient selection criteria, posing a limitation to this review. The existing trials differ significantly in inclusion criteria, scoring systems, surgical techniques, and use of maintenance therapy.”
#7 I believe it should be highlighted in the conclusion section that SCS should be provided in the context of MDT decision making as previously mentioned in the discussion section
Reply
We agree with the Reviewer. The final sentence in the conclusion has been revised to the following:
“In conclusion, this SR suggests patients with relapsed platinum-sensitive epithelial ovarian cancer may benefit from complete or optimal cytoreductive surgery, and thus should be evaluated for surgical treatment in the context of a multidisciplinary team decision making based on their functional status and disease burden.”
